# Chemical Composition, Phytotoxic, Antimicrobial and Insecticidal Activity of the Essential Oils of *Dracocephalum integrifolium*

**DOI:** 10.3390/toxins11100598

**Published:** 2019-10-13

**Authors:** Shixing Zhou, Caixia Wei, Chi Zhang, Caixia Han, Nigora Kuchkarova, Hua Shao

**Affiliations:** 1CAS Key Laboratory of Biogeography and Bioresource in Arid Land, Xinjiang Institute of Ecology and Geography, Urumqi 830011, China; zhoushixing16@mails.ucas.ac.cn (S.Z.); weicaixia16@mails.ucas.ac.cn (C.W.); cxhani@ms.xjb.ac.cn (C.H.); nkn.bell.ixrv@mail.ru (N.K.); 2University of Chinese Academy of Sciences, Beijing 100049, China; 3Shandong Provincial Key Laboratory of Water and Soil Conservation and Environmental Protection, College of Resources and Environment, Linyi University, Linyi 276000, China; zc@ms.xjb.ac.cn

**Keywords:** *Dracocephalum integrifolium* Bunge, essential oil, sabinene, phytotoxic activity, antimicrobial activity, insecticidal activity

## Abstract

The present investigation studied the chemical composition of the essential oils extracted from *Dracocephalum integrifolium* Bunge growing in three different localities in northwest China and evaluated the phytotoxic, antimicrobial and insecticidal activities of the essential oils as well as their major constituents, i.e., sabinene and eucalyptol. GC/MS analysis revealed the presence of 21–24 compounds in the essential oils, representing 94.17–97.71% of the entire oils. Monoterpenes were the most abundant substances, accounting for 85.30–93.61% of the oils; among them, sabinene (7.35–14.0%) and eucalyptol (53.56–76.11%) were dominant in all three oils, which occupied 67.56–83.46% of the total oils. In general, phytotoxic bioassays indicated that the IC_50_ values of the oils and their major constituents were below 2 μL/mL (1.739–1.886 mg/mL) against *Amaranthus retroflexus* and *Poa annua*. Disc diffusion method demonstrated that the oils and their major constituents possessed antimicrobial activity against *Bacillus subtilis*, *Pseudomonas aeruginosa*, *Escherichia coli*, *Saccharomyces cerevisiae*, and *Candida albicans*, with MIC values ranging from 5–40 μL/mL (4.347–37.712 mg/mL). The oils, sabinene and eucalyptol also exhibited significant pesticidal activity, with the mortality rates of *Aphis pomi* reaching 100% after exposing to 10 μL oil/petri dish (8.694–9.428 mg/petri dish) for 24 h. To the best of our knowledge, this is the first report on the chemical composition, phytotoxic, antimicrobial and insecticidal activity of the essential oils extracted from *D. integrifolium*; it is noteworthy to mention that this is also the first report on the phytotoxicity of one of the major constituents, sabinene. Our results imply that *D. integrifolium* oils and sabinene have the potential value of being further exploited as natural pesticides.

## 1. Introduction

Yield losses caused by agricultural pests including arthropods, diseases, and weeds are estimated to account for about 35% in major crops worldwide [1]. Synthetic chemical pesticides are widely accepted in prevention of diseases and insect pests in crop growth; however, extensive application of synthetic chemicals has led to the development of resistance in insects and weeds [2,3]. Besides, wide use of synthetic pesticides has also caused many problems including the acute and chronic toxicity to human and mammals, as well as adverse and residual effects on the environment [4].

Traditionally, numerous plant products have a record of safe application in various industries [5]. Plant derived substances, including essential oils, have been an increasing interest in search of alternatives to chemical pesticides because of their innate biodegrade ability, minimal effects on non-target organisms and the environment [6,7]. In fact, essential oils as well as oil constituents/derivatives have successfully been commercialized as environment friendly pesticides such as clove oil, eucalyptol and so on [8,9].

The genus *Dracocephalum* (family Labiates) consists of around 60 annual or perennial herbs that distributes widely in Southern Europe and temperate Asia, with 32 species and 7 varieties can be found growing in China [10,11]. A literature survey of the chemical composition of essential oils produced by *Dracocephalum* plants revealed the presence of remarkable variations among their major constituents [12]. These oils were found to possess various biological activities such as antimicrobial [13,14,15,16,17,18,19,20], antioxidant [13,17,19,21,22,23], insecticidal [24,25], phytotoxic [26,27], cytotoxic [17], antispasmodic [28] and antinociceptive [29] activity.

Among various *Dracocephalum* species, *D. integrifolium* Bunge is an aromatic plant that has been used traditionally as an important medicinal herb known as “Marzan Juxi” in Uighur in treating cough and asthmadistributes in Central Asia [30]. Phytochemical study indicated that this plant mainly contains flavonoids, volatile oils, lactones and other compounds [11]. So far, there are no reports on the chemical composition and biological activities of the essential oil of *D. integrifolium*. In this study, we aim to: (i) analyze the phytochemical profile of the essential oils extracted from *D. integrifolium* growing in 3 different localities in northwest China; (ii) evaluate the phytotoxic, antimicrobial and pesticidal activities of these essential oils as well as their major constituents. The possibility of utilizing the essential oils and their major constituents as environment friendly pesticides is also discussed.

## 2. Results

### 2.1. Chemical Composition of the Essential Oils

The yield of the essential oils obtained from stems and leaves of *D. integrifolium* harvested from three different localities was 0.17% (DI1), 0.16% (DI2), and 0.17% (DI3). In total 24, 21 and 21 compounds were identified from DI1, DI2 and DI3 essential oils, representing 97.71%, 95.71%, and 94.17% of the entire oil, respectively (Table 1). Sabinene and eucalyptol were the most abundant constituents in all 3 oils, with the content of sabinene ranging from 7.35% to 14.0%, and eucalyptol ranging from 53.56% to 76.11%; these 2 compounds occupied 67.56%, 83.46%, 75.66% of the DI1, DI2 and DI3 oils, respectively (Figure 1, Figure 2 and Figure 3). Monoterpenes were overwhelmingly dominant, accounting for 85.30%, 93.61%, 86.12% of the DI1, DI2 and DI3 oils. The following compounds, α-thujene, α-pinene, sabinene, α-terpinene, eucalyptol, β-ocimene, γ-terpinene, L-pinocarveol, terpinen-4-ol, myrtenal and caryophyllene oxide can be found in all 3 oils, whereas o-cymene, 3-carene, β-elemene, aromadendrene, β-copaene, cis-α-bisabolene, β-bisabolene, nerolidol and α-bisabolol were unique in DI1 oil, isopinocamphone, α-terpineol, myrtenol and epizonarene were found only in DI2 oil, and β-thujene, (+)-ledol were present only in DI3 oil.

### 2.2. Phytotoxic Activity of the Essential Oils and their Major Constituents

Phytotoxic activity of the essential oils (concentrations tested ranged from 0.125 to 5 μL/mL, 0.109–4.714 mg/mL) as well as their major constituents was determined by comparing their plant regulatory effect on shoot and root length of two receiver plants, i.e., *A. retroflexus* and *P. annua*. At the lowest concentration tested (0.125 μL/mL, 0.91–0.938 mg/mL), all 3 oils significantly promoted root elongation of *A. retroflexus*; however, sabinene, eucalyptol and their mixture did not exert any promotive activity. With the increase of concentration, inhibitory effects were observed when oils were applied starting from 0.5–1 μL/mL (0.455–0.938 mg/mL), whereas the effective dose for sabinene, eucalyptol and their mixture was 2 μL/mL (1.739 mg/mL), 0.25 μL/mL (0.236 mg/mL), and 0.5 μL/mL (0.457 mg/mL), respectively. In fact, 2 μL/mL (1.739–1.886 mg/mL) oils or the major constituents resulted in over 50% reduction on root length: DI1, DI2, DI3 oils inhibited root elongation of *A. retroflexus* by 86.5%, 100%, 100%, meanwhile sabinene, eucalyptol and their mixture exerted 66.7%, 63.0%, and 76.0% reduction on root length of *A. retroflexus*, respectively. At the highest concentration tested (5 μL/mL, 4.347–4.714 mg/mL), seed germination of *A. retroflexus* was markedly inhibited by the oils and the major constituents. Shoot growth responded similarly to the oils and the major constituents but to a lesser extent (Figure 4 and Figure 5).

Similarly, DI1 and DI2 oils significantly stimulated root growth of the other tested species, the monocot plant *P. annua*; DI3 oil, on the contrary, exhibited obvious phytotoxic activity starting from the lowest concentration tested, 0.125 μL/mL (0.117 mg/mL). For sabinene, eucalyptol and their mixture, significant plant growth suppressive effect can be detected starting from 0.25–0.5 μL/mL (0.217–0.471 mg/mL). Like *A. retroflexus*, over 50% reduction on root length was triggered by 2 μL/mL (1.739–1.886 mg/mL) oils or the major constituents: the inhibition rates of 3 oils on root length of *P. annua* were 65.8% (DI1), 100% (DI2) and 80.5% (DI3), respectively, whilst at the same concentration, seedling height were reduced to 61%, 100% and 65.3% of the control, respectively. When the concentration reached 5 μL/mL (4.347–4.714 mg/mL), seed germination of *P. annua* was basically completely prohibited (Figure 6 and Figure 7).

Strength of the phytotoxic activity was compared by calculating the IC_50_ values for the oils and the major constituents (Figure 8). Among the three oils, DI3 showed the most potent phytotoxic activity on root growth of both receiver species, whereas DI1 exhibited the weakest activity. Slight promotive activity of the mixture of sabinene and eucalyptol was observed on root elongation of *A. retroflesus* and shoot growth of both test species.

### 2.3. Antimicrobial Activity of the Essential Oils and Their Major Constituents

Disc diffusion method demonstrated that the essential oils and their major constituents had suppressive effect on tested microorganisms. In general, DI2 oil showed the strongest activity against the tested strains compared with other oils; meanwhile, the major constituents were also very effective; for example, sabinene resulted in the largest inhibition zone (6.63 cm) in the assay (Table 2).

The minimal inhibitory concentration (MIC) of *D. integrifolium* oils and their major constituents was also evaluated in order to assess the strength of their antimicrobial activity (Table 3). *B. subtillis* turned out to be the most sensitive microorganism with the lowest MIC values (5 μL/mL, 4.347–4.714 mg/mL) for the oils and the major constituents; in comparison, *S. cerevisiae* was the most tolerant strain, with MIC values of 40 μL/ mL for sabinene (34.776 mg/mL) and the mixture of sabinene and eucalyptol (36.576 mg/mL).

### 2.4. Pesticidal Activity of Essential Oils and their Major Constituents

The insecticidal effect of *D. integrifolium* oils against *A. pomi* was summarized in Figure 9, Figure 10, Figure 11, Figure 12, Figure 13 and Figure 14. Results showed that all three essential oils as well as the major constituents posed toxic effect on *A. pomi*, and higher mortality was observed as the doses of essential oils and exposure period increased. After 48 h exposure to the oils at 2 μL/petri dish (1.820–1.876 mg/petri dish), the mortality rates of *A. pomi* were 56.0%, 72.6% and 78.3%, respectively; and when the concentration reached 10 μL (9.100–9.380 mg), the mortality rates were 100% for all 3 oils after 24 h exposure. In order to compare the strength of the oils, LC_50_ and LC_90_ values were calculated and the results were as follows: LC_50_ values were 3.103 (2.824 mg) and 0.975 (0.887mg) μL/petri dish after 24h, 48h of exposure for DI1 oil, 1.621 (1.499 mg) and 0.702 (0.649 mg) μL/petri dish for DI2 oil, and 1.953 (1.832 mg) and 0.652 (0.612 mg) μL/petri dish for DI3 oil, respectively, whereas the LC_90_ values after 24 h and 48 h exposure were 30.826(28.03 mg) and 6.231(5.67mg) μL/petri dish, 8.32 (7.696 mg) and 4.604 (4.259 mg) μL/petri dish, 8.185 (7.678 mg) and 3.115 (2.922 mg) μL/petri dish, respectively. In general, DI2 and DI3 oils had stronger effect against *A. pomi* compared with DI1 oil.

Sabinene, eucalyptol and their mixture also exhibited strong insecticidal effect. Among them, sabinene showed the most potent effect on *A. pomi* with LC_50_ and LC_90_ values of 0.667 (0.579 mg) and 1.86 (1.617 mg) μL/petri dish after 24 h exposure, and 2 μL/petri dish (1.739 mg/petri dish) treatment resulted in 100% motality. The mixture of sabinene and eucalyptol affected *A. pomi* with LC_50_ and LC_90_ values of 0.262 (0.239 mg) and 1.161 (1.062 mg) μL/petri dish after 48 h exposure. When the concentration reached 5 μL (4.347–4.714 mg), the motality rate of eucalyptol, sabinene and their mixture reached 100% after 48 h exposure. All the compounds can kill all of test insects after 24 h at the dose of 10 μL/petri dish (8.694–9.428 mg/petri dish).

## 3. Discussion

The chemical composition of essential oils produced by a number of *Dracocephalum* species have been reported, especially *D. moldavica* and *D. kotschyi*, mostly because of their important value as traditional medical herbs [19,20,21,27,28,31,32]. Suleimen et al. [26] investigated the chemical composition of *D. peregrinum* and detected 1, 8-cineol (synonym for eucalyptol, 18.5%), α-pinene (8.4%), limonene (5.8%), β-caryophyllene oxide (5.5%), and spathulenol (3.3%) as the major constituents; however, the major constituents of some other *Dracocephalum* species differ greatly from *D. integrifolium*. Mahmood et al. [12] found citronellol (74.9%), Citronellal (6.7%) and Citronellyl formate (6.7%) were abundant in *D. heterophyllum* oil; Nezhadali et al. [18] reported that the major constituents of the essential oil obtained from *D. subcapitatum* were geranial (63.4%), limonene (23.4%), p-Menth-1-en-9-ol (4.4%) and caryophyllene E (4.3%). Various factors such as geographic locality and surrounding climate, species variety or ecotype, seasonal variations, stress, post-harvest processing, etc. might influence the chemical profile of essential oils [33] In terms of *D. integrifolium*, VOCs released by powdered dry plant materials were studied by Liu et al. [30] using headspace solid-phase microextraction (HS-SPME) combined with gas chromatography-mass spectrometry (GC-MS) method, and eucalyptol and cymene were discovered as the most abundant ingredients. To the best of our knowledge, our study is the first to report the chemical profile of the essential oil of *D. integrifolium*, with eucalyptol and sabinene being the dominant compounds in the oils; both compounds are common constituents of plant essential oils and have been demonstrated to attribute to certain biological activities of the oils.

Essential oils are produced by plants from the family Compositae, Umbelliferae, etc., that function as pollinator attractants, determinants of vegetation patterning or regulatory factor of community structure via allelopathy [34]. Further study revealed that essential oils possess various biological activities such as antioxidant, cytotoxic, anti-inflammatory, anti-microbial activity, etc. [35,36]. In terms of the phytotoxic activity of essential oils produced by *Dracocephalum* species, there was one single report on the significant allelopathic activity of the essential oil produced by a new chemotype of *D. kotschyi* that was characterized with abundant limonene-10-al and limonene [27]. *D. integrifolium* has the ability to emit phytotoxic volatile compounds into the environment, indicating the possible involvement of allelopathy in its competition against neighboring plants. Allelopathy refers to any direct and indirect harmful or beneficial effect by one plant on another through the production of chemical compounds that are released into environment, which is often found to contribute to successful establishment of dominance in a certain habitat [37,38]. Meanwhile, it is noteworthy to mention that the mechanism of allelopathy is rather complicated and may be very difficult to be confirmed, mainly because of the uncertainty of the fate of the potential volatile allelochemicals after they are released into the environment. On the one hand, these compounds might act directly as phytotoxins to affect other plants’ growth if they can accumulate to effective doses in the soil; on the other hand, they might indirectly impact on neighbor’s growth by altering the soil’s physico-chemical properties as well as the community structure and function of soil biota [39,40].

In most cases biological activities of a particular essential oil is determined by one or two of its major components, however synergistic effect may occur in other circumstances [33,41]. Essential oil constituents such as eucalyptol, camphor, β-myrcene, limonene, thymol, geraniol, α- and β-pinene, bornyl acetate, citronellal, menthol, borneol and a-terpineol, etc. have been found to exhibit plant growth inhibitory activity [42,43]. Eucalyptol and sabinene are common components present in a number of essential oils [44,45,46,47,48,49]. So far, there is no report on the phytotoxicity of sabinene, therefore we are the first to study this biological activity of this compound; however, there are some essential oils with phytotoxic activity have sabinene as the major constituents, implying that sabinene might play an important role as the active compound in the oils. For instance, the essential oil produced by *Ravensara aromatic* that was rich in sabinene (the sabinene chemotype) caused 52.5% mortality of rice seedlings at a dose of 100 μL/L, and completely killed the garden cress seedlings at a dose of only 5 µl/L [46]. The essential oil of *Ferulago angulate* also exhibited considerable phytotoxic activity, with sabinene (6.89%) being one of its major constituents [50]. *Zingiber montanum* oil reduced germination and inhibited seedling growth of lettuce significantly, which is abundant with sabinene 13.5–38.0% [51]. Eucalyptol is also known as 1, 8-cineole, which has been previouslyt demonstrated to be phytotoxic [52,53,54,55]; this compound was considered an active constituent of cyanobacterial VOCs that negatively affect other algae’s growth by reducing photosynthetic abilities and inducing degradation of photosynthetic pigments [35,56]; Barton et al. [53] found that eucalyptol posed post-emergence herbicidal activity against annual ryegrass and radish in a dose-dependent manner, and suppression of seedling growth was significant at and above 0.1 mol/L on radish, meanwhile application of eucapyptol prohibited seedling growth of ryegrass when the concentration was above 0.1 mol/L, with root suppression first occurring at 0.0316 mol/L and shoot suppression at 0.1 mol/L. Martino et al. [43] compared the antigerminative activity of 27 monoterpenes and found that eucalyptol suppressed radicle elongation of both radish and garden cress in a significant way at the lowest concentration tested (10^−6^ M).

Phytotoxicity of the essential oils and their major ingredients may lead to the discovery of environment-friendly natural herbicides. For example, the high effectiveness of eucalyptol and its analogs led to the successful commerciliazation of a herbicide, cinmethylin, which was designed based on the chemical structure of eucalyptol, as the active ingredient 1, 4-cineole is a 2-benzyl ether-substituted analog of eucalyptol [9]. Another herbicide, Burnout II (Bonide Products Inc., New York) used clove oil (common volatile oil constituent) as the major component in the ingredients [8].

Essential oils have been reported to possess antimicrobial activity, including a number of *Dracocephalum* species [15,19,20,57]. Essential oils produced by two *Dracocephalum* species (*D. polychaetum* Bornm. and *D. surmandinum* Rech.f.) exhibited significant antibacterial activity against both gram-positive and gram-negative bacteria [17]. Essential oil obtained from another *Dracocephalum* species, *D. foetidumby*, also showed inhibitory activity against most of the tested pathogenic bacteria and yeast strains with MIC values ranging from 26 to 2592 μg/mL; furthermore, the oil was found to exhibit significant antimicrobial activity against methicilin-resistant *Staphylococcus aureus* (MRSA) strains, indicating this oil has potential medical value [20]. Farimani et al. [14] examined the antimicrobial activity of *D. kotschyi* essential oils against gram-positive (*Staphylococcus aureus*) and gram-negative (*Escherichia coli*) bacteria and found *S. aureus* was the most sensitive strain with an IC_50_ value of 2 mg/mL. Zhang et al. [19] tested the antimicrobial activity of the essential oil of *D. heterophyllum* against nine bacterial strains, one yeast, three fungi, and found that MIC values for bacteria, yeast, and fungi strains were 0.039–0.156, 0.156, and 0.313–2.500 mg/mL, respectively. Besides frequently reported antibacterial effect, the essential oil of *D. heterophyllum* also showed promising antifungal activity against Colletotrichum species [15].

Antimicrobial activity of essential oils could be attributed to the high content of certain components; for instance, the high antimicrobial activity of *Thymus pulegioides* oil was caused by one the its abundant constituents, thymol [58,59]. Essential oils rich in eucalyptol are reported to possess antimicrobial activity; eucalyptol was found to exhibit inhibitory activity against various microorganisms such as *E. coli*, *S. aureus*, *B. cereus*, *P. aeruginosa*, etc. [60]. Trinh et al. [61] found that eucalyptol significantly decreased the amount of viable *Gardnerella vaginalis* and *Candida albicans* in the vaginal cavity and myeloperoxidase activity in mouse vaginal tissues compared with controls. Vimal et al. [62] discovered that eucalyptol and sabinene could be potent inhibitors of salmonella target protein L–asparaginase, and sabinene was even better than the standard drug, Ciprofloxacin and the natural substrate L-asparagine. Some essential oils that are rich in sabinene were found to possess antimicrobial activity: *Artemisia kulbadica* oil (sabinene 25.1%) showed inhibitory activity against six bacterial strains and one fungal [63]; *Alpinia nutans* oil (sabinene 27.8%) was effective against *Pasteurella multocida, Escherichia coli, Salmonella enterica, Shigella fluxneri and Staphylococcus aureus* [64]; *Laserpitium latifolium* oil (sabinene 26.8%) inhibited growth of *Staphylococcus aureus, Staphylococcus epidermidis, Micrococcus luteus* [65]. The effectiveness of *D. integrifolium* oil and its major constituents implies their potential value to be further explored as natural antimicrobial agents.

Essential oils of *Dracocephalum* plants have been demonstrated to have insecticidal activity. *D. Kotschyi* essential oil was effective against green peach aphids, and *D. ruyschiana*, *D. foetidum*, *D. moldavica*, *D. fruticztlosum* and *D. peregrinum* essential oils had good insecticidal effect on mosquito larvae [24,27]; Chu et al. [25] found *D. moldavica* essential oil exhibited strong fumigant toxicity against *Sitophilus zeamais* and *Tribolium castaneum* adults with LC_50_ values of 2.65 and 0.88 mg/L, respectively. The two major components of the essential oils, sabinene and eucalyptol, also have insecticidal activity. Sukontason et al. [66] studied the effects of eucalyptol on house fly (*Musca domestica*) and blow fly (*Chrysomya megacephala*) and found for *M. domestica*, males were more susceptible than females, with the LD_50_ being 118 and 177 µg/fly, respectively; as of *C. megacephala*, the LD_50_ values were 197 µg/fly for males and 221 µg/fly for females. Another research showed that eucalyptol possessed strong fumigant toxicity against two insects, *Tribolium castaneum* (LC_50_ = 5.47 mg/L air) and *Lasioderma serricorne* (LC_50_ = 5.18 mg/L air; [67]). Another major constituent, sabinene, was also reported to be insecticidal. Wang et al. [68] found sabinene was strongly repellent against *Tribolium castaneum* with the LC_50_ value of 18.2 mg/L of air, although its effect was weaker compared with the positive control, DEET (N, N-diethyl-3-methylbenzamide). Sabinene also exhibited strong fumigant toxicity against the maize weevils with LC_50_ value of 9.12 mg/L [69]. In another study, sabinene possessed noticeable repellent activity against adults of the granary weevil (*Sitophilus granaries*) at low dosages [70]. Therefore, the dominant components of *D. integrifolium*, eucalyptol and sabinene, are quite likely the main responsible active ingredients contributing to the oils’ insecticidal activity.

## 4. Conclusions

Plant derived essential oils have the potential to be further explored as agricultural chemicals to control pathogenic fungi, pests and weeds to decrease the negative impact of synthetic agents due to the fact that they are effective, selective, biodegradable, and less toxic to the environment. To the best of our knowledge, this is the first report on the chemical composition, phytotoxic, antimicrobial and pesticidal activities of the essential oil extracted from *D. integrifolium*; this is also the first report on the phytotoxic activity of one of the oil’s major constituents, sabinene. Our results imply that *D. integrifolium* oil and its major constituents are valuable candidates to be further explored as biopesticides.

## 5. Materials and Methods

### 5.1. Plant Materials

Stems and leaves of *D. integrifolium* were collected from 3 localities in northern Tianshan Mountain region located in Xinjiang Uyghur Autonomous Region, China, in June 2018. Specimens were identified by Prof. Wenjun Li from Xinjiang Institute of Ecology and Geography, Chinese Academy of Sciences. Voucher specimens were deposited with the serial numbers of XJBI018112, XJBI018113, and XJBI018114 at the herbarium of Xinjiang Institute of Ecology and Geography, Chinese Academy of Sciences. Plant materials were collected from 3 different localities in Xinjiang Uyghur Autonomous Region, China, including DI1 (Lat 43°96′80′′ N, Lon 87°16′48′′E, alpine pasture, with an elevation of 1683m), DI2 (Lat 43°96′80′′N, Lon 85°87′63′′E, riverside, with an elevation of 1146 m), and DI3 (Lat 43°86′25′′N, Lon 86°20′53′′E, valley, with an elevation of 1462 m).

### 5.2. Essential oil Extraction

Three hundred fresh stems and leaves of *D. integrifolium* were cut into fragments to extract essential oil using a Clevenger-type apparatus. The hydrodistillation procedure lasted for 4 h, followed by collection of the oil; the oil was then dried over anhydrous sodium sulfate and kept in a sealed vial at 4 °C until required. Materials collected from 3 sites were extracted and stored separately. The following formula was used to calculate the yield of *D. integrifolium* oil:Oil yield (%, *V*/*W*) = volume of essential oils (mL)/fresh weight of plant material (g) × 100.

### 5.3. GC/MS Analysis of the Essential Oils

The phytochemical profile of the oil was analyzed using a 7890A/5975C GC/MS system (Agilent Technologies, Palo Alto, CA, USA) equipped with FID and a DB-5MS 5% Phenyl Methyl Silox column (30 m × 0.25 mm; 0.25 μm film thickness). The carrier gas was helium (flow rate: 1 mL/min). The oven temperature was held at 60 °C for 5 min and then programmed from 60 °C to 280 °C at 3 °C/min. Mass spectra were taken at 70 eV with mass range from m/z 40–800 amu. The temperature of both injector port and detector port was kept at 280 °C; sample injection volume, 0.1μL; split ratio was 50: 1. Relative amounts of individual compound were calculated based on GC peak areas without FID response factor correction. Identification of single component was determined by comparison of their mass spectra and retention indices (RI, calculated by linear interpolation relative to retention times of a standard mixture of C7–C40 n-alkanes) with the data given in NIST (National Institute of Standards and Technology), and published literature [52,71].

### 5.4. Phytotoxic Activity of the Essential Oils and Their Major Constituents

The major constituents of the essential oil, i.e., sabinene and eucalyptol, were purchased from Sigma-Aldrich Co. (St. Louis, USA). Phytotoxic effect of the essential oils, their major constituents (sabinene and eucalyptol), along with their mixture prepared at 10.5: 65.1 (sabinene: eucalyptol; the average ratio of sabinene and eucalyptol in the oils) was evaluated by conducting bioassays against a dicot plant, *Amaranthus retroflexus* L., and a monocot plant, *Poa annua* L. Seeds were surface sterilized with 0.5% HgCl_2_ before use. The essential oils, sabinene, eucalyptol, and mixture of sabinene and eucalyptol were diluted with distilled H_2_O to obtain 0.125 (0.109–0.118 mg), 0.25 (0.217–0.236 mg), 0.5 (0.435–0.471 mg), 1 (0.869–0.943 mg), 2 (1.739–1.886 mg), 5 (4.347–4.714 mg) μL/mL solutions, with acetone (final volume < 0.5%) as the initial solvent; a preliminary experiment demonstrated that acetone at such concentration did not significantly influence seedling growth of receiver plants. Five mL of 0.5% acetone in distilled H_2_O (control) or diluted solutions (treatments) were added to each petri dish (9 cm in diameter, lined with a single layer of Whatman No. 2 filter paper), followed by sowing of 10 seeds of receiver plants. Petri dishes were sealed with parafilm to keep the moisture and avoid oil evaporation. An incubator was used to store all the petri dishes in the dark at 25 °C. Seedlings were observed and measured after 5 days of cultivation for the dicot plant *A. retroflexus*, and 7 days for the monocot plant *P. annua.* Five replicates were made for all phytotoxic bioassays (in total 50 seedlings were measured).

### 5.5. Antimicrobial Effect of the Essential Oils and Their Major Constituents

#### 5.5.1. Strains

Five microorganisms, including *Bacillus subtilis* (ATCC6633), *Pseudomonas aeruginosa* (ATCC27853), *Escherichia coli* (ATCC25922), *Saccharomyces cerevisiae* (ATCC9763), and *Candida albicans* (ATCC10231) were chosen to evaluate the antimicrobial effect of the essential oils as well as their major constituents; all strains were obtained from China General Microbiological Culture Collection Center (CGMCC). Among the test strains, *B. subtilis* is a gram positive bacterium; *P. aeruginosa* and *E. coli* are gram negative bacteria; *S. cerevisiae* and *C. albicans* are fungi. Antimicrobial activity of the essential oils and their major constituents were tested using disc diffusion method (for measurement of radius of the inhibition zone) and tube dilution method (for measurement of MIC) [72,73].

#### 5.5.2. Determination of Radius of Inhibition Zone

Bacteria or fungi suspensions were adjusted to 1 × 10 ^7^ CFU/mL and spread on LB medium (for bacteria) or PDA medium (for fungi) using sterile glasses spreader. Subsequently, filter paper discs (5 mm in diameter, Whatman No. 2) that were impregnated with 20 μL (17.388–18.856 mg) of essential oils or the major constituents were placed on LB or PDA medium. Negative controls were prepared with distilled water. Petri dishes were then incubated at 37 °C for 24 h for bacteria, and 48 h for fungi. All assays were performed in triplicates. Antimicrobial activity was evaluated by measuring the radius of the inhibition zones to the nearest millimeter.

#### 5.5.3. Determination of MIC

The essential oils and their major constituents were diluted with LB medium (for bacteria) and PDB medium (for fungi) supplemented with 0.5% of Tween-20 to give oil concentrations of 5 (4.347–4.714 mg), 10 (8.694–9.428 mg), 20 (17.388–18.856 mg), 40 (34.776–37.712 mg) μL/mL. Samples were first sterilized with a 0.22 μm Millipore before application, and then inoculated with 50 μL of fresh culture of the test microorganisms at 1 × 10^7^ CFU/mL and incubated at 37 °C for 24 h for bacteria, and 48 h for fungi. MIC was defined as the lowest concentration of oils that had no macroscopically visible growth. All assays were performed in triplicates.

### 5.6. Insecticidal Bioassay

*Aphis pomi* was used to determine the insecticidal activity of the oils and their major constituents. Whatman No. 2 filter paper discs (1 cm × 1 cm) were soaked in *D. integrifolium* oils, sabinene, eucalyptol, and the mixture of sabinene and eucalyptol at the following concentrations: 0, 0.25 (0.217–0.236 mg), 0.5 (0.435–0.471 mg), 1 (0.869–0.943 mg), 2 (1.739–1.886 mg), 5 (4.347–4.714 mg) and 10 (8.694–9.428 mg) μL/disc. Paper discs were then placed onto the inner side of the lid of each petri dish (9 cm in diameter) using adhesive tape. This procedure was adopted in order to prevent direct contact between the oils and the insects. Treatments consisted of 20 adult aphids, which were placed onto fresh apple tree leaves that were kept on a layer of moist filter paper. Petri dishes were then covered and transferred to an incubator and kept at 25 ± 2 °C with a photoperiod L/D = 16:8 for 2 days. Each treatment was replicated three times (n = 60).

### 5.7. Statistical Analysis

The significance of the phytotoxic/antimicrobial/pesticidal activity was first examined by ANOVA (*p* < 0.05) and then analyzed using Fisher’s LSD test at *p* < 0.05 level. All of the statistical analyses were performed using SPSS 13.0 software package.

## Figures and Tables

**Figure 1 toxins-11-00598-f001:**
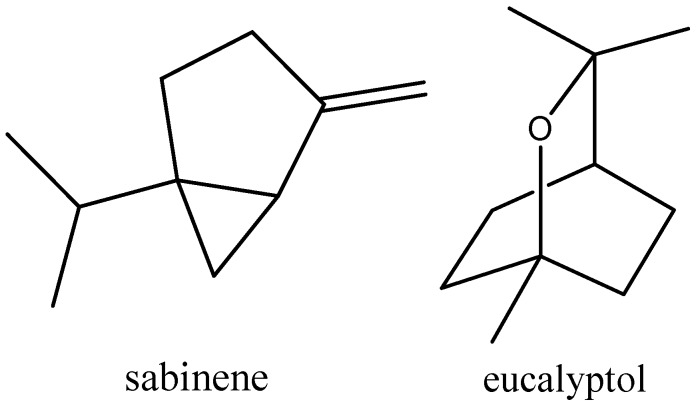
Chemical structures of sabinene and eucalyptol.

**Figure 2 toxins-11-00598-f002:**
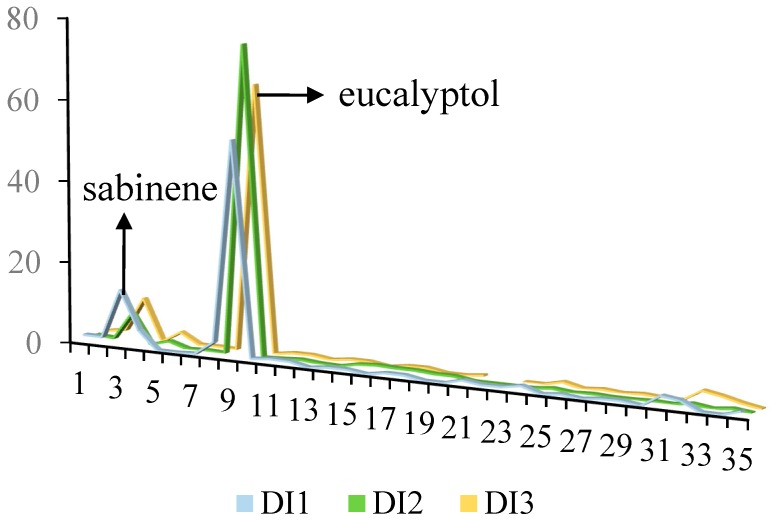
Comparison of the chemical composition of *D. integrifolium* essential oils collected from three different localities.

**Figure 3 toxins-11-00598-f003:**
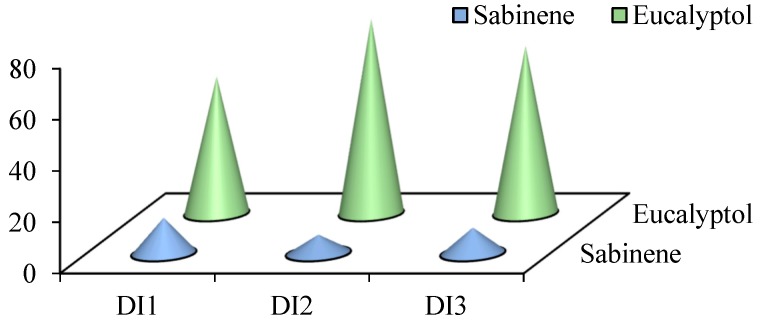
Comparison of the percentage of the major constituents of *D. integrifolium* essential oils collected from 3 different localities.

**Figure 4 toxins-11-00598-f004:**
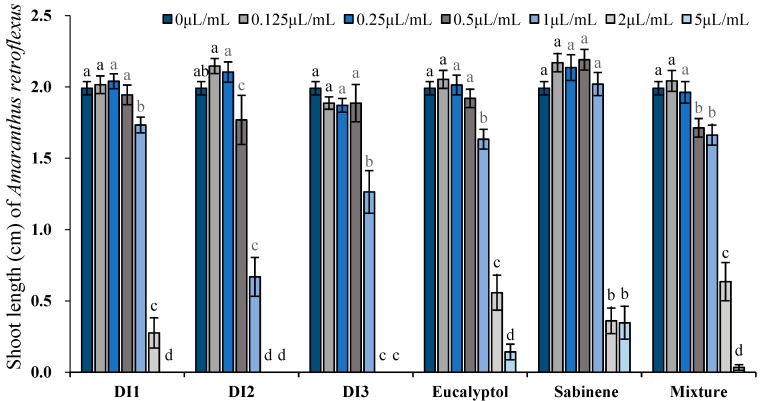
Phytotoxic effects of the essential oils of *D. integrifolium* and their major constituents, sabinene, eucalyptol, and their mixture on root growth of *A. retroflexus* examined by ANOVA (*p* < 0.05) and analyzed using Fisher’s LSD test at *p* < 0.05 level. Each value is the mean of five replicates ± SE (n = 50). Means with different letters indicate significant differences at *p* < 0.05 level according to Fisher’s LSD test.

**Figure 5 toxins-11-00598-f005:**
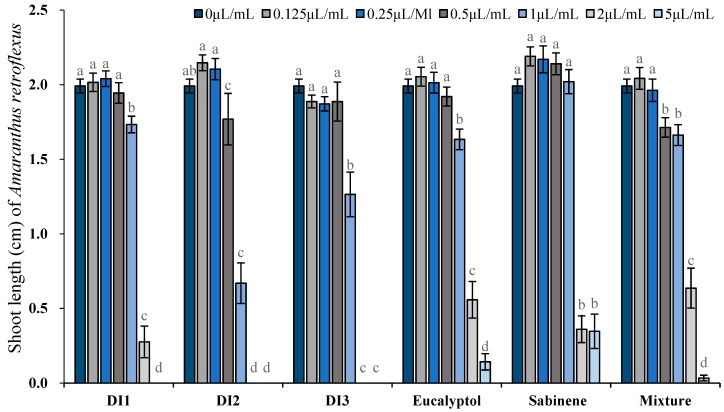
Phytotoxic effects of the essential oils of *D. integrifolium* and their major constituents, sabinene, eucalyptol, and their mixture on shoot growth of *A. retroflexus* examined by ANOVA (*p* < 0.05) and analyzed using Fisher’s LSD test at *p* < 0.05 level. Each value is the mean of five replicates ± SE (n = 50). Means with different letters indicate significant differences at *p* < 0.05 level according to Fisher’s LSD test.

**Figure 6 toxins-11-00598-f006:**
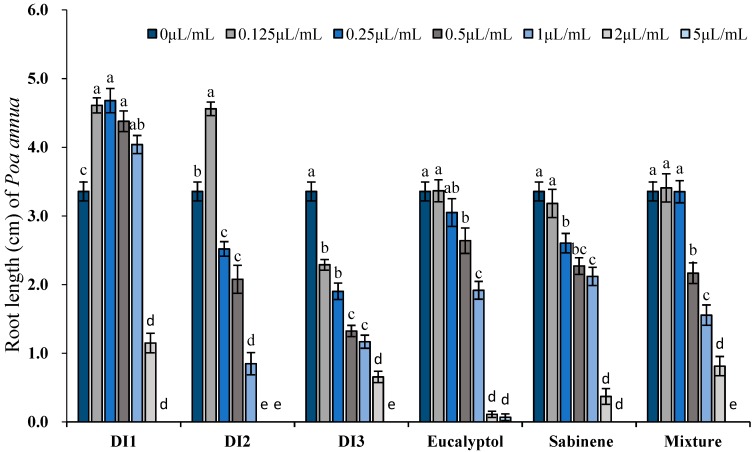
Phytotoxic effects of the essential oils of *D. integrifolium* and their major constituents, sabinene, eucalyptol, and their mixture on root growth of *P. annua* examined by ANOVA (*p* < 0.05) and analyzed using Fisher’s LSD test at *p* < 0.05 level. Each value is the mean of five replicates ± SE (n = 50). Means with different letters indicate significant differences at *p* < 0.05 level according to Fisher’s LSD test.

**Figure 7 toxins-11-00598-f007:**
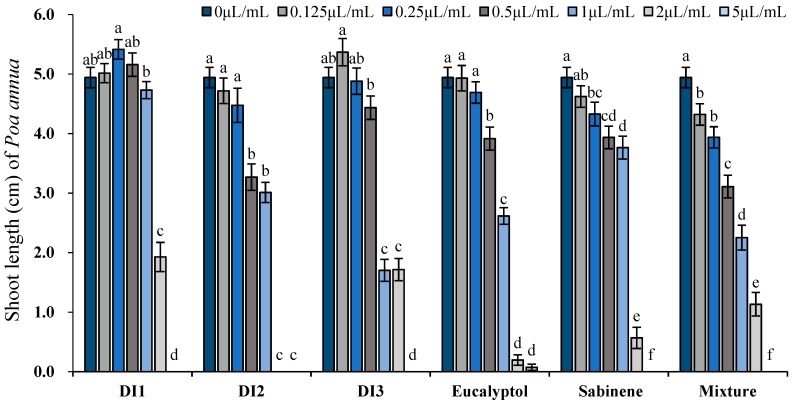
Phytotoxic effects of the essential oils of *D. integrifolium* and their major constituents, sabinene, eucalyptol, and their mixture on shoot growth of *P. annua* examined by ANOVA (*p* < 0.05) and analyzed using Fisher’s LSD test at *p* < 0.05 level. Each value is the mean of five replicates ± SE (n = 50). Means with different letters indicate significant differences at *p* < 0.05 level according to Fisher’s LSD test.

**Figure 8 toxins-11-00598-f008:**
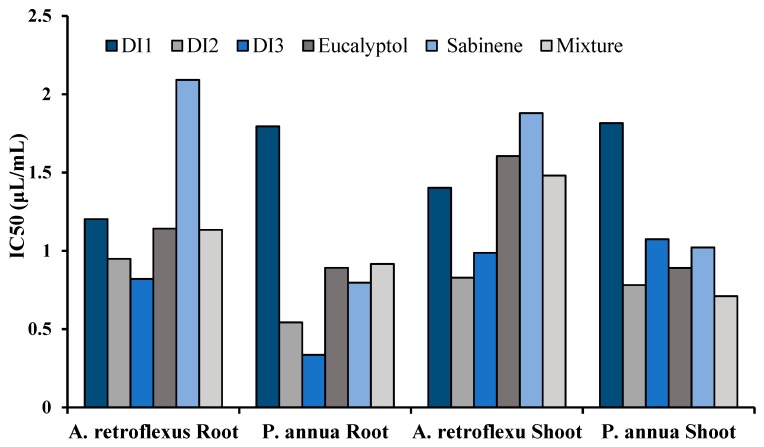
IC_50_ values of *D. integrifolium* essential oils and their major constituents, sabinene, eucalyptol, and their mixture on root and shoot length of *A. retroflexus* and *P. annua.*

**Figure 9 toxins-11-00598-f009:**
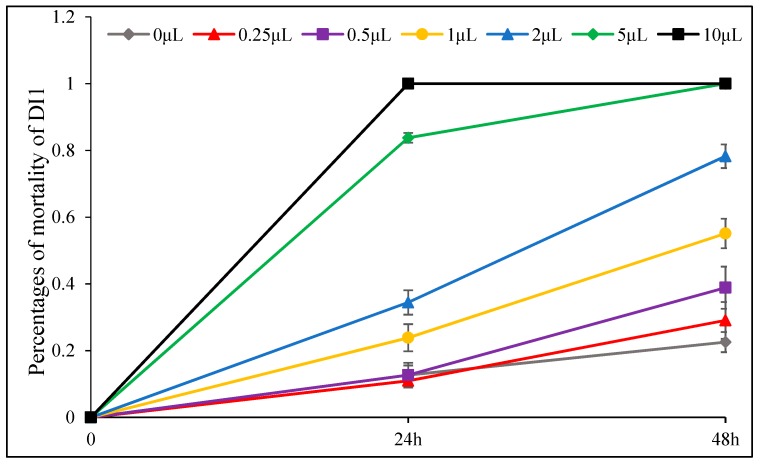
Percent mortality of *A. pomi* adults after treatment with essential oil of DI1 according to doses exposure and treatment times.

**Figure 10 toxins-11-00598-f010:**
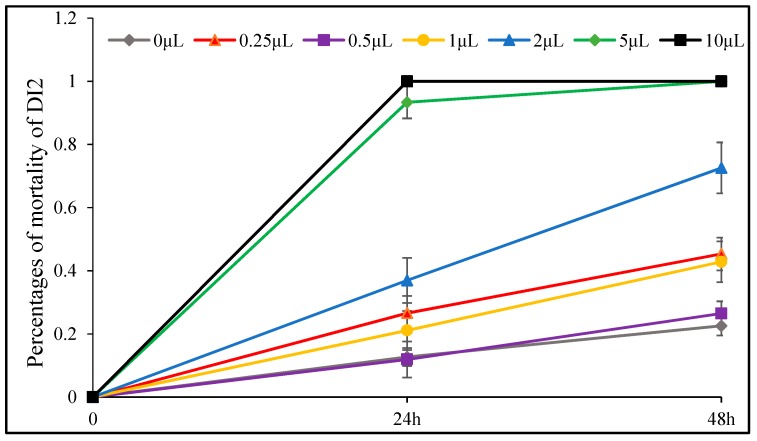
Percent mortality of *A. pomi* adults after treatment with essential oil of DI2 according to doses exposure and treatment times.

**Figure 11 toxins-11-00598-f011:**
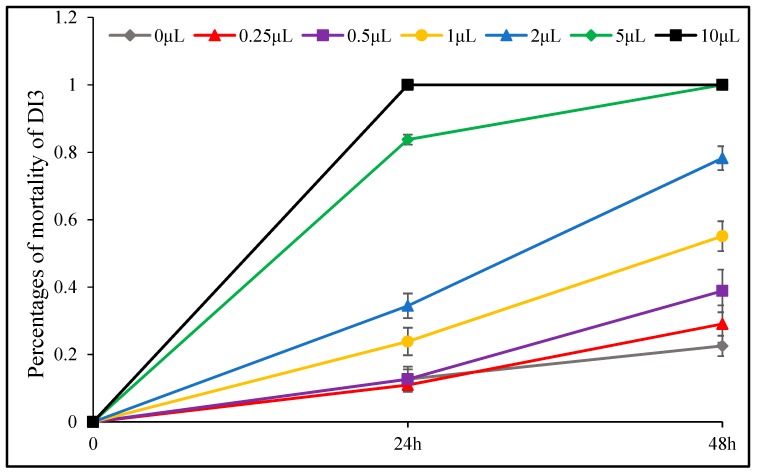
Percent mortality of *A. pomi* adults after treatment with essential oil of DI3 according to doses exposure and treatment times.

**Figure 12 toxins-11-00598-f012:**
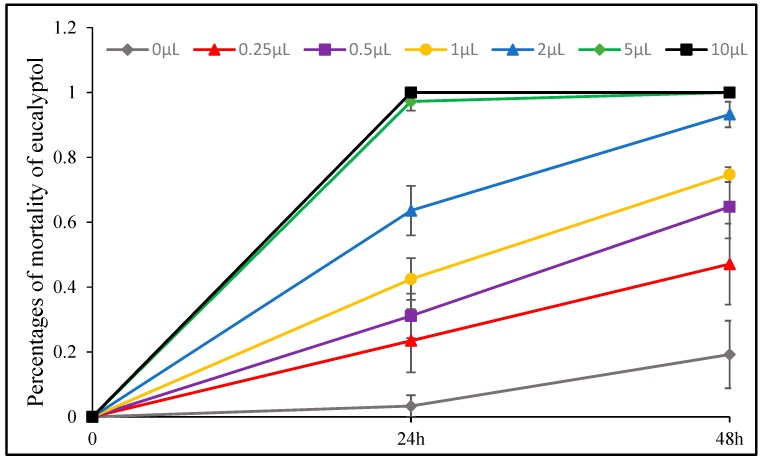
Percent mortality of *A. pomi* adults after treatment with essential oil of eucalyptol according to doses exposure and treatment times.

**Figure 13 toxins-11-00598-f013:**
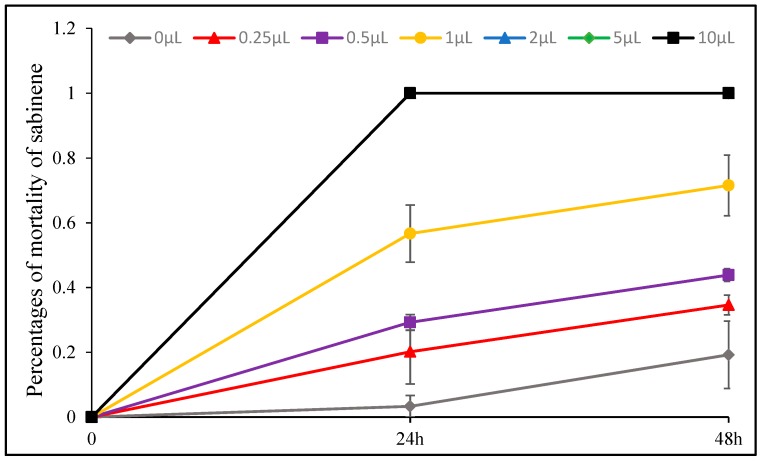
Percent mortality of *A. pomi* adults after treatment with sabinene according to doses exposure and treatment times.

**Figure 14 toxins-11-00598-f014:**
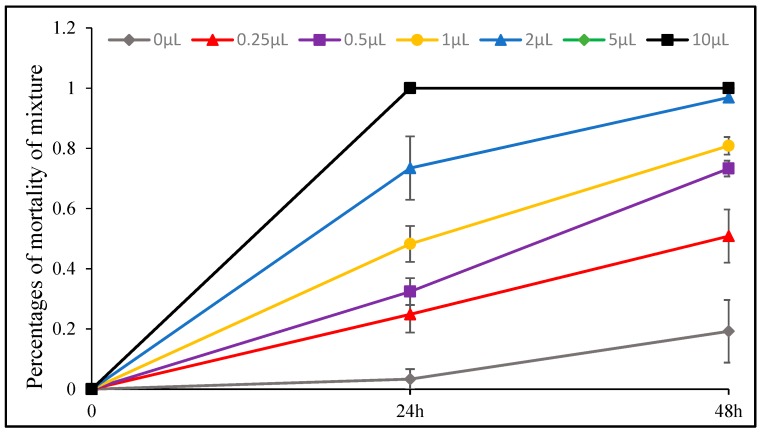
Percent mortality of *A. pomi* adults after treatment with of eucalyptol and sabinene mixture (65.1: 10.5) according to doses exposure and treatment times.

**Table 1 toxins-11-00598-t001:** Chemical composition of the essential oils obtained from *D. integrifolium* plants growing in three different localities.

Serial No.	Compound Name	RI^a^	RI^b^	Area (%)			
				DI1	DI2	DI3	Average
1	α-Thujene	913	924	1.87	0.98	1.07	1.31
2	α-Pinene	918	935	1.84	0.67	1.83	1.45
3	Sabinene	958	954	14	7.35	10.11	10.49
4	β-Myrcene	968	969	4.49	0.11	-	2.30
5	β-Pinene	976	975	-	1.4	2.76	2.08
6	β-Thujene	996	978	-	-	0.17	0.17
7	α-Terpinene	1002	1016	0.28	0.21	0.23	0.24
8	o-Cymene	1006	1018	3.66	-	-	3.66
9	Eucalyptol	1013	1030	53.56	76.11	65.55	65.07
10	3-Carene	1022	1031	0.61	-	-	0.61
11	β-Ocimene	1033	1035	1.09	0.23	0.7	0.67
12	γ-Terpinene	1041	1056	1.01	0.53	0.73	0.76
13	Terpinolene	1070	1079	-	0.13	0.1	0.12
14	Linalool	1080	1098	0.61	-	0.71	0.66
15	L-Pinocarveol	1110	1139	0.62	0.97	0.72	0.77
16	Isopinocamphone	1137	1162	-	1.27	-	1.27
17	Terpinen-4-ol	1151	1177	0.84	1.08	0.68	0.87
18	Myrtenal	1156	1190	0.82	1.09	0.76	0.89
19	α-Terpineol	1163	1198	-	0.75	-	0.75
20	Myrtenol	1169	1206	-	0.73	-	0.73
21	α-Copaene	1359	1372	1.24	-	0.56	0.90
22	β-elemene	1373	1392	0.61	-		0.61
23	Aromadendrene	1397	1419	0.7	-	-	0.70
24	β-copaene	1456	1459	1.51	-	-	1.51
25	Germacrene D	1456	1451	-	0.49	1.09	0.79
26	cis-α-Bisabolene	1483	1495	0.54	-	-	0.54
27	γ-Muurolene	1489	1477	-	0.22	0.39	0.31
28	β-Bisabolene	1489	1505	0.55	-	-	0.55
29	δ-Cadinene	1498	1519	0.63	-	0.35	0.49
30	Epizonarene	1498	1495	-	0.2	-	0.20
31	Nerolidol	1540	1554	2.94	-	-	2.94
32	Caryophyllene oxide	1546	1543	2.27	0.64	2.83	1.91
33	(+)-ledol	1559	1550	-	-	1.99	1.99
34	Cubenol	1582	1601	-	0.55	0.84	0.70
35	α-Bisabolol	1657	1691	1.42	-	-	1.42
	Monoterpene hydrocarbons			28.85	11.61	17.7	19.39
	Oxygenated monoterpenes			56.45	82	68.42	68.96
	Sesquiterpene hydrocarbons			5.78	0.91	2.39	3.03
	Oxygenated sesquiterpenes			6.63	1.19	5.66	4.49
	Total identified			97.71	95.71	94.17	95.86
	Numbers of compounds			24	21	21	22.00
	Yield (%, V/W)			0.17	0.16	0.17	0.17

RI^a^ Retention index measured relative to *n*-alkanes (C_7_–C_40_) using DB-5MS column. RI^b^ Retention index from literature.

**Table 2 toxins-11-00598-t002:** Mean radius of inhibition zones (cm) of the essential oils and their major constituents tested against five microbial stains.

Essential Oils/ Major Constituents	*E. coli*	*B. subtillis*	*P. aeruginosa*	*S. cerevisiae*	*C. albicans*
DI1	1.08 ± 0.08	2.13 ± 0.43	1.30 ± 0.10	2.75 ± 0.40	2.55 ± 0.15
DI2	1.55 ± 0.05	2.30 ± 0.10	1.85 ± 0.05	3.83 ± 0.13	2.90 ± 0.05
DI3	1.15 ± 0.05	2.15 ± 0.00	1.65 ± 0.05	3.53 ± 0.28	3.05 ± 0.05
Sabinene	1.15 ± 0.05	3.08 ± 0.08	1.58 ± 0.03	6.63 ± 0.03	3.35 ± 0.05
Eucalyptol	1.78 ± 0.38	3.03 ± 0.08	1.18 ± 0.03	1.95 ± 0.05	2.03 ± 0.23
Mixture	1.83 ± 0.03	2.10 ± 0.00	1.55 ± 0.05	3.60 ± 0.00	2.48 ± 0.18

**Table 3 toxins-11-00598-t003:** Minimum inhibitory consentration (MIC) of the essential oils and their major constituents tested against 5 microbial stains.

Essential Oils/ Major Constituents	*E. coli*	*B. subtillis*	*P. aeruginosa*	*S. cerevisiae*	*C. albicans*
DI1	10	5	10	15	10
DI2	10	5	10	10	10
DI3	15	5	15	15	5
Eucalyptol	15	5	10	20	10
Sabinene	20	5	10	40	15
Mixture	10	5	15	40	10

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
