# Peer review of "Chemical Composition, Phytotoxic, Antimicrobial and Insecticidal Activity of the Essential Oils of Dracocephalum integrifolium"

_toxins, 2019, doi:10.3390/toxins11100598_

Round 1
Reviewer 1 Report
The authors extracted essential oils from Dracocephalum integrifolium Bunge that grew at three different localities in northwest China and evaluated their bioactivities and their chemical composition. The authors found that the oils were predominantly composed of eucalyptol and sabinene. Bioassays reveled high toxicity against weeds, bacteria and yeasts, and insects.
I think the study was well-conducted, the methods are sound, and that there are no major experimental flaws. The only weakness is as follows:
The authors present the concentrations of essential oils used for the bioassays in μl/ml. The data should be presented in molarity. This way, one can directly compare the biological effects of eucalyptol and sabinene in relative terms. This will also help to put the results of this study into a broader context since most studies report the concentrations in μg/ml, which, although is commonly used, is not fully correct, but it is more informative than μl/ml.Author Response
Please see the attachment.

Reviewer 2 Report
This is an interesting study about the chemical composition, phytotoxic, antimicrobial and insecticidal activity of the essential oils of Dracocephalum integrifolium.
The paper is well written and the presentation quality is good. However, a major revision of the methods employed in the study, and consequently of the results, is necessary.
In detail:
1) review all names to write in italics throughout the text.
2) correct Candida albicans (no Canidia, no alicans!) throughout the text, including the tables.
3) Material and methods
a) Rewrite the paragraph 5.5.2. The disc diffusion test is not suitable to evaluate the antifungal activity. In literature you can find several works where the method most frequently used is described. Moreover, you impregnated the discs with 20 μl of each essential oil: if the EO is not diluted, the amount is very high.
b) Revise the paragraph 5.5.3. You used oil concentrations of 5, 10, 20, 40 μL/mL. Express the concentration in mg/ml.
4) Rewrite the results in relation to the "new" methods.
Round 2
Reviewer 2 Report
Dear Authors,
the quality of your manuscript has been significantly improved. The manuscript is worthy to be published in the current form.